# Diagnostic Challenges in the Pathological Approach to Pleural Mesothelioma

**DOI:** 10.3390/cancers17030481

**Published:** 2025-02-01

**Authors:** Stefano Lucà, Giovanna Pignata, Alessandro Cioce, Cecilia Salzillo, Rossella De Cecio, Gerardo Ferrara, Carminia Maria Della Corte, Floriana Morgillo, Alfonso Fiorelli, Marco Montella, Renato Franco

**Affiliations:** 1Pathology Unit, Department of Mental and Physical Health and Preventive Medicine, University of Campania Luigi Vanvitelli, 80138 Naples, Italy; stefano.luca@unicampania.it (S.L.); alessandro.cioce@unicampania.it (A.C.); marco.montella@unicampania.it (M.M.); 2Department of Experimental Medicine, PhD Course in Public Health, University of Campania Luigi Vanvitelli, 80138 Naples, Italy; cecilia.salzillo@unicampania.it; 3Istituto Nazionale Tumori IRCCS Fondazione G. Pascale, 80131 Naples, Italy; giovanna.pignata@istitutotumori.na.it (G.P.); r.dececio@istitutotumori.na.it (R.D.C.); gerardo.ferrara@istitutotumori.na.it (G.F.); 4Department of Precision and Regenerative Medicine and Ionian Area, Pathology Unit, University of Bari Aldo Moro, 70121 Bari, Italy; 5Department of Precision Medicine, Medical Oncology, University of Campania Luigi Vanvitelli, 80138 Naples, Italy; carminiamaria.dellacorte@unicampania.it (C.M.D.C.); floriana.morgillo@unicampania.it (F.M.); 6Thoracic Surgery Unit, Department of Translational Medicine, University of Campania Luigi Vanvitelli, 80138 Naples, Italy; alfonso.fiorelli@unicampania.it

**Keywords:** pleural mesothelioma, malignant, histopathology, molecular tests

## Abstract

Malignant pleural mesothelioma poses a significant diagnostic challenge for pathologists, requiring a comprehensive multidisciplinary approach. Diagnosis relies on morphological, immunohistochemical, and sometimes molecular assessments, integrated with clinical and radiological findings. The first diagnostic challenge is distinguishing MPM from metastatic pleural lesions or benign mesothelial proliferations. Immunohistochemical markers such as podoplanin, WT1, calretinin, and HEG1 are pivotal, though none is entirely specific. In the same way, BAP1 and MTAP provide enhanced sensitivity and specificity for differentiating malignant from benign conditions and, in more complex cases, molecular tests can detect valuable genetic alterations. The International Mesothelioma Interest Group regularly updates its guidelines to optimize and refine diagnostic processes. The aim is to enhance diagnostic precision and improve the clinical management of MPM.

## 1. Introduction

Malignant pleural mesothelioma (MPM) is a rare and very aggressive neoplasm arising from mesothelial cells lining the pleural cavity [1,2].

The diagnosis of MPM is often challenging and typically requires a multidisciplinary approach. It relies on compatible morphological and immunohistochemical findings obtained from an adequate tissue sample, usually a biopsy, and less frequently from an effusion or exfoliative cytology specimen. These findings must be assessed within an appropriate clinical and radiological context. In some cases, molecular studies may be necessary to resolve diagnostic uncertainty. Importantly, a history of asbestos exposure should not influence the pathological evaluation when confirming or excluding mesothelioma [3].

Several factors make the diagnosis of MPM a persistent challenge for pathologists. Firstly, the rarity of this neoplasm, combined with the shortage of pathologists and particularly the lack of professionals specialized in thoracic pathology in many centers [4,5]. Indeed, distinguishing MPM from several neoplastic and non-neoplastic entities presents a significant challenge, which is often exacerbated by the small size of biopsy samples. This limitation increasingly necessitates the use of multiple ancillary techniques to achieve an accurate diagnosis [6,7,8].

The morphological and immunohistochemical criteria to distinguish MPM from reactive pleural disease and non-mesothelial pleural neoplasms (primary or metastatic) have recently emerged, leading to a multitude of new concepts with highly relevant diagnostic implications. In this context, the International Mesothelioma Interest Group (iMig) periodically publishes the guidelines for pathologic diagnosis of mesothelioma, with the last update in 2023, to guide and facilitate the diagnosis [3].

## 2. Histopathology General Features

According to the current WHO Classification of Thoracic Tumours [9], mesothelial neoplasms are classified into benign tumors and mesothelioma. MPM is subclassified into three distinct major histological subtypes:Pleural epithelioid mesothelioma (PEM)Pleural sarcomatoid mesothelioma (PSM)Pleural biphasic mesothelioma (PBM)

All are characterized by specific histopathological, biological, and prognostic features [9].

### 2.1. Pleural Epithelioid Mesothelioma: A Histopathological Overview

PEM is the most frequent histologic subtype of MPM (60–70% of diagnosed cases) and exhibits established morphological heterogeneity with several reported architectural, cytological, and stromal features that are prognostically significant [10,11]. PEM most often includes bland, polygonal, oval, or cuboidal cells with eosinophilic cytoplasm and round nuclei, with vesicular chromatin and central nucleoli. Mitoses may be observed. However, some cases display pronounced cytological atypia (coarse chromatin and central prominent nucleoli) combined with high mitotic count and necrosis [8].

Five architectural patterns with defined prognostic implications are recognized:Tubulopapillary: a combination of tubules and papillae with connective tissue cores and/or clefts lined by a range of bland to more atypical cuboidal cells.Trabecular: small uniform cells forming thin cords or, sometimes, a single-file arrangement.Adenomatoid: structures with lace-like or signet ring appearance.Micropapillary: papillary structures lacking a fibrovascular core.Solid: sheets of cohesive tumor cells.

Several unconventional cytological features have also been described; in detail:Rhabdoid cytomorphology: discohesive cells with abundant eosinophilic cytoplasm, an eccentric nucleus with a prominent nucleolus, and a rounded eosinophilic cytoplasmic inclusion that sometimes causes nuclear indentation [12,13].Deciduoid cytomorphology: cells with abundant eosinophilic cytoplasm, resembling deciduoid cells of pregnancy [14].Small cell cytomorphology: solid nests of closely packed small cells exhibiting a well-defined cell membrane and high nuclear cytoplasmic ratio [15,16].Clear cell cytomorphology: large cells with clear cytoplasm and round central nuclei [17].Signet ring cytomorphology: optically clear cytoplasmic vacuoles pushing the nucleus to one side within tumor cells [18].Lymphohistiocytoid cytomorphology: polygonal tumor cells morphologically like histiocytes mixed with a marked lymphoid infiltrate [19,20,21,22].Pleomorphic cytomorphology: tumor cells with prominent anaplastic nuclei as well as bizarre nuclei, often also with multinucleated tumor giant cells [23,24].

All these cytological variants are unusual, but they should be recognized to avoid misdiagnosis with some metastatic carcinomas mimics, such as small cell carcinoma of the lung (SCLC), clear cell renal cell carcinoma (CCRCC), other clear cell carcinomas, or adenocarcinoma with signet ring cell features.

The stromal background also shows variability. It can be fibrous, exhibiting varying degrees of cellularity, ranging from hyalinized acellular to highly cellular, or myxoid. Pronounced myxoid changes can be appreciated in a minority of cases, with isolated cells or clusters of cytologically bland, often vacuolated, epithelioid cells suspended in an Alcian Blue positive hyaluronate matrix [25,26].

All the reported histopathological features of PEM are summarized in Table 1 and should be described in the pathological report as they show critical prognostic implications. Tubulopapillary, trabecular, and adenomatoid patterns correlate with a better prognosis, just like lymphohistiocytoid cytomorphology and myxoid stroma. Solid and micropapillary growth patterns and high-grade cytological features (rhabdoid and pleomorphic) are conversely associated with worse prognosis [1].

The current WHO Classification of Thoracic Tumours [9] proposes a two-tiered histopathological grading system (low and high grade) for PEM. This system is based on two morphological features: nuclear grade (defined by the combination of mitotic index and nuclear atypia) and presence/absence of necrosis (Table 2). Areas showing the highest-grade features should be used to assign the tumor grade. This parameter should be routinely reported in pathological reports of PEM because this grading system has been demonstrated to be strongly predictive of survival in patients with PEM and it allows identification of neoplasms with more aggressive behavior [9].

The diagnosis of PEM can be straightforward, but sometimes it is very challenging. The main entities to distinguish are pleural metastatic carcinomas from other sites (mainly lung carcinomas) and obviously mesothelial proliferation, both benign (adenomatoid tumor of the pleura, well-differentiated papillary mesothelial tumors) (Table 3) and reactive (reactive mesothelial hyperplasia).

### 2.2. Pleural Sarcomatoid Mesothelioma: A Histopathological Overview

PSM is the second most common and the most aggressive histological type of MPM (5–15% of all MPMs) [27,28]. Its morphologic features include a range of growth patterns as well as variable cell morphology. Usually, PSM consists of elongated and tapered spindle cells, arranged in a fascicular or haphazard growth, with various degrees of nuclear atypia, ranging from relatively bland to highly atypical and pleomorphic nuclei with very prominent nucleoli. Necrosis is common and a high mitotic index, also with atypical forms, could be seen. Most commonly, a fibrosarcomatous growth pattern is encountered [29,30]. However, lymphohistiocytoid, pleomorphic, and transitional subtypes are recognized as cytological variants of PSM.

The transitional variant displays sheets of large, elongated, and plump cells with a well-defined cellular border, abundant cytoplasm, high nucleocytoplasmic ratio, and prominent nucleoli. These cells have an intermediate morphology between epithelioid and sarcomatoid types because they start to lose their epithelioid cellular structure but are more round than sarcomatoid cells without a spindle-shaped form and lacking frank sarcomatous features and, at the same time, they are more discohesive than epithelioid cells and reticulin stain highlights single cells, just like in PSM [31,32].The pleomorphic variant exhibits large, atypical, and pleomorphic cells, including multinucleated and giant elements. The cells show irregular, markedly atypical, and hyperchromatic nuclei. A high mitotic count, also with bizarre mitoses, is a frequent feature [24].

Heterologous differentiation such as osteosarcomatous, rhabdomyosarcomatous, and chondrosarcomatous elements can be observed in some cases of PSM [33]. Finally, pleural desmoplastic mesothelioma (PDM) is now considered a variant of PSM and represents an extraordinarily difficult diagnostic hurdle. PDM shows spindle cells with minimal atypia arranged haphazardly in a so-called “patternless pattern” with a dense hyalinized stroma that resembles pleural hyaline plaque [34,35]. All the reported histopathological features of PSM are summarized in Table 4.

PSM and PDM represent an extremely complex diagnostic challenge and constitute a diagnosis of exclusion. The main entities to distinguish are thoracic sarcomas, sarcomatoid carcinomas (mainly pleomorphic carcinoma of the lung), and chronic fibrosing pleuritis.

### 2.3. Pleural Biphasic Mesothelioma: A Histopathological Overview

PBM accounts for 15% to 30% of MPMs and it is composed of both epithelioid and sarcomatoid components. Its diagnosis is independent of the percentage of the two components in small biopsy samples. However, in definitive resection specimens, at least 10% of each component is required to establish a diagnosis of PBM. Nevertheless, some studies have suggested a prognostic cut-off for the sarcomatoid component ranging between 50% and 80% [32]. The prognosis of PBM is considered intermediate, falling between that of pure PEM and pure PSM [36].

Even though the data are limited, some evidence suggests that PBM with a predominance of sarcomatoid morphology (>50%) may have a poorer prognosis. Therefore, the relative percentage of both the epithelioid and sarcomatoid components should be carefully reported for biopsy and resection samples [32,37].

## 3. A Preinvasive Lesion: The Concept of Mesothelioma In Situ (MIS)

Mesothelioma in situ (MIS) was first described by Simon et al. [38] who identified a preinvasive lesion in a series of invasive MPM cases. However, specific diagnostic criteria for MIS have only been established in recent years, primarily due to the rarity of this preinvasive lesion [39,40,41].

MIS is a preinvasive single-layer surface proliferation of neoplastic cells. It is composed of flat or cuboidal cells with or without cytologic atypia and may show small or complex papillary proliferations and/or small surface nodules (Figure 1) [3].

This entity cannot be diagnosed based only on morphological criteria but it requires the demonstration of loss of BAP1 nuclear expression by immunohistochemistry (Figure 1) and/or *CDKN2A* homozygous deletion, identified either by FISH or by loss of MTAP cytoplasmic expression by immunohistochemistry, using a validated assay [42,43].

## 4. The Pivotal Role of Immunohistochemistry: Past and Future

The diagnosis of malignant pleural mesothelioma (MPM) is straightforward when typical clinico-radiologic and morphological features are present. However, in some cases, reaching an accurate diagnosis of MPM can pose one of the most challenging difficulties in routine surgical pathology practice [44]. The first question to ask is:Is the pleural lesion a mesothelial or non-mesothelial proliferation?

Or

Is the mesothelial proliferation benign or malignant in nature?

The morphological resemblance between MPM and several epithelial and non-epithelial neoplasms explains the first question. Notably, carcinoma represents the most common morphological differential diagnosis for MPM. Additionally, the significant morphological overlap between benign mesothelial proliferations and MPM can hinder a reliable distinction based on histomorphology alone. Therefore, in this context, ancillary diagnostic tests, particularly immunohistochemical staining, have become crucial for achieving an accurate diagnosis of mesothelioma [45].

However, there are three major MPM histologic subtypes and the immunophenotype is generally different between each subtype. Moreover, immunohistochemical biomarkers currently used in clinical practice are not entirely sensitive and specific [46].

### 4.1. Is the Pleural Lesion a Mesothelial or Non-Mesothelial Proliferation?

The first question to answer is “Is the pleural lesion a mesothelial or non-mesothelial proliferation?”. Metastases to the pleura occur more often than primary pleural tumor and they may clinically present with pleural effusions, pleural thickening and pleural-based masses, just like primary pleural neoplasms or reactive pleural processes [47].

#### 4.1.1. Pleural Epithelioid Mesothelioma, What Differential Diagnoses?

The diagnosis of PEM is sometimes straightforward, but it can be challenging due to its ability to mimic many other epithelial neoplasms. Malignancies from any primary neoplasm may metastasize to the pleura but primary lung carcinomas are undoubtedly the first differential diagnosis [3,48].

Other common neoplasms to be considered in the differential diagnosis are those of the breast [49]. In this setting, an accurate and selected panel of immunomarkers allows establishment of the histological lineage in all the cases with sufficient tissue for evaluation. Broad-spectrum cytokeratins are virtually always expressed by PEM, with a diagnostic sensitivity of 100%. If PEM is morphologically suspected but cytokeratins are negative, alternative diagnoses should be considered, such as melanoma, epithelioid vascular neoplasms, or lymphoma. The best known and most used mesothelial markers are podoplanin (D2-40), Wilms tumor-1 (WT1), calretinin, and CK5/6. These biomarkers have an excellent diagnostic sensitivity, especially for the epithelioid histotype of MPM, which is obviously different from biomarker to biomarker, but none of them shows a diagnostic specificity of 100% for mesothelial origin [45,50].

Podoplanin (D2-40): a membranous expression pattern supports mesothelial origin and has highest sensitivity for pleural mesothelioma, showing a membranous staining in 80–100% of PEMs (Figure 2) [3,45,51,52,53,54,55]. However, its specificity for the mesothelial lineage is not absolute. Although podoplanin (D2-40) is essentially not expressed in lung adenocarcinomas (3% of focally positive lung adenocarcinomas) [3,56], it is positive in 60% of lung squamous cell carcinomas [3,57,58,59,60].WT1: a nuclear expression pattern supports mesothelial origin, while cytoplasmic staining should be disregarded (Figure 2). WT1 is a very sensitive biomarker, showing an expression ranging from 70–100% in PEM [3,45,55,58,61,62]. However, although it is not absolute, WT1 shows the highest specificity among common mesothelial biomarkers, showing lack of expression in lung adenocarcinomas [3,45,56] and renal cell carcinomas [3,45,63], virtually no expression in lung squamous cell carcinomas (only 0–2% of reported positive cases) [3,45,57], and a very low expression in breast carcinomas (5–8% of reported positive cases) [3,45,54].Calretinin: a combined cytoplasmic and nuclear expression pattern supports mesothelial origin, while other staining patterns, especially lack of nuclear staining, should be disregarded. This biomarker shows high sensitivity but moderate specificity for mesothelial lineages and, in detail, calretinin is 80–100% sensitive for PEM but it is also expressed in other neoplasms, such as lung squamous cell carcinomas (expression reported in approximately 35–40% of cases), lung adenocarcinomas (expression reported in approximately 0–10% of cases), breast carcinomas (expression reported in approximately 15% of cases and up to 38% of triple-negative breast carcinomas), and renal cell carcinomas (expression reported in approximately 0–10% of cases). All these data demonstrate that calretinin is not a perfect biomarker for mesothelial origin and should be evaluated as part of a broader immunohistochemical panel [3,64].Cytokeratin 5/6 (CK 5/6): a cytoplasmic expression pattern supports mesothelial origin, but it shows an intermediate sensitivity (ranging from 51% to 100%) and specificity. CK5/6 is usually expressed in lung squamous cell carcinomas (95–100% of the cases) and breast carcinomas (5% of the cases). On the other hand, this biomarker is particularly valuable in the differential diagnosis with lung adenocarcinoma, of which less than 5% are positive, so its use is typically limited to cases where lung adenocarcinoma is essentially the only other possible diagnosis [3].

In our routine practice, we always use a panel of at least 2/3 biomarkers of mesothelial lineage, choosing the combination of podoplanin (D2-40) and WT1 primarily and possibly adding calretinin. On the other hand, the diagnostic panel should always be completed with two broad-spectrum carcinoma biomarkers at least, such as claudin-4, Ber-EP4, or MOC-31.

Claudin-4: the best epithelial biomarker currently used in routine diagnostic practice [3,65,66,67,68]. A membranous expression pattern supports epithelial lineages with a diagnostic sensitivity and specificity of 92–100% and 94–100%, respectively (Figure 2) [54,58,65,66,67,69,70,71,72]. Only rare PEMs show claudin-4 expression but with a characteristic staining pattern, represented by a focal (<10% of tumor cells) and granular/dot-like cytoplasmic stain [58]. In summary, claudin-4 is a great epithelial biomarker, and it should be used routinely, and especially in challenging cases, to exclude an epithelial origin of the neoplasm.Ber-EP4: an antibody against the epithelial cell adhesion molecular (Ep-CAM) expressed on epithelia and in various carcinomas. A strong and diffuse membranous staining supports epithelial lineage. A specificity ranging from 60% to 100% in the differential diagnosis with mesothelial neoplasms was reported, with cellular expression observed in about 10% of PEMs [52,56,57,70,73,74,75].MOC-31: an antibody against the epithelial cell adhesion molecular (Ep-CAM) expressed on epithelia and in various carcinomas, just like Ber-EP4, but it shows a higher diagnostic sensitivity for epithelial neoplasms compared to the latter [76]. A 90–100% diagnostic specificity for epithelial lineage was reported [69,70,77] with a strong and diffuse membranous staining [74]. However, Chapel et al. [45] reported some mesothelioma cases with a strong and diffuse expression of MOC-31 and/or Ber-EP4 and, therefore, they suggest always combining one of these antibodies with an immunohistochemical panel that includes claudin-4 and two mesothelial lineage biomarkers.CEA: an older epithelial origin biomarker potentially employed as a valid alternative to the previous ones [78]. With CEA monoclonal antibody, a diffuse cytoplasmic staining with membrane enhancement supports an epithelial lineage [79]. It shows a good but variable sensitivity (expression rates of 84% in lung adenocarcinoma and 92% in lung squamous cell carcinoma) and an equally good specificity, being positive in <5% of PEMs with a typically focal expression [3]. It must be considered that polyclonal antibodies do not show diagnostic accuracy comparable to monoclonal antibodies. However, a broader immunohistochemical panel is always recommended in this diagnostical setting.

#### 4.1.2. Pleural Sarcomatoid Mesothelioma, What Differential Diagnoses?

The differential diagnosis of PSM from other sarcomatoid tumors involving the pleura is critical for optimal clinical management but it can be highly challenging for surgical pathologists. Sarcomas showing spindle cell or pleomorphic morphology and pleomorphic carcinomas of the lung (PCLs), especially the spindle cell variant, represent the main clinical entities to differentiate from PSM [80]. The immunohistochemical workup of PSM is different and typically more extensive than that of PEM [81]. The immunohistochemical panel should include, in addition to cytokeratins and mesothelial markers, a series of mesenchymal markers, such as desmin, S-100, myogenin, STAT6, SS18-SSX, CD34, ERG, CD31, and FLI1, and melanoma markers, like SOX10, HMB45, and Melan A. The choice of the most appropriate panel obviously depends on the morphological features of the lesion in order to avoid unnecessary and sometimes confusing immunostaining. Carcinoma markers, such as claudin-4, MOC-31, and Ber-EP4 are not usually helpful in the differential diagnosis, so they should not be included in the panel [3]. Primarily, keep in mind that cytokeratins are undoubtedly the most useful biomarkers in this diagnostic setting. PSM is usually at least focally positive for cytokeratin. The best choice is a broad-spectrum cytokeratin but, if a single non-broad-spectrum cytokeratin must be used, cytokeratin 8/18 (CK8/18) could be positive in tumors in which other cytokeratins are negative.

Broad-spectrum cytokeratin (pan-cytokeratin, cytokeratin AE1/AE3, CAM 5.2): cytoplasmic reactivity is highly sensitive for mesothelioma, including PSM. Almost all PSMs exhibit immunoreactivity for at least one cytokeratin, at least focally, and the proportion increases if a cytokeratin cocktail is used (90% sensitivity) (Figure 3). So, cytokeratin positivity is extremely useful in excluding spindle cell sarcoma or melanoma and in confirming the mesothelial nature of the proliferation, although areas of heterologous differentiation in PSM are often cytokeratin negative. This immunoreactivity should rule out most but not all sarcomas. In cases with inconclusive immunohistochemistry, some molecular tests for sarcoma and other mesenchymal neoplasms can be used. Noteworthily, broad-spectrum cytokeratins do not allow differentiation between PSM and sarcomatoid carcinoma, especially PCL.

Sarcomatoid carcinoma, especially PCL, represents an important and complex differential diagnosis. The most used mesothelial markers have limited diagnostic sensitivity for PSM. In detail:Podoplanin (D2-40): a membranous expression pattern supports mesothelial origin (Figure 3). This biomarker shows the highest sensitivity for the diagnosis of PSM among the main mesothelial biomarkers, even if the reported diagnostic sensitivity values are variable, ranging from 50–60%, 75–90%, and even up to 100% [82,83,84,85,86]. However, it should be kept in mind that podoplanin (D2-40) is the most sensitive biomarker for PSM and that the staining may be focal. On the other hand, podoplanin (D2-40) expression has been reported in about 25–30% of PCLs [51,87,88] and some germinal and mesenchymal neoplasms [89].Calretinin: a combined cytoplasmic and nuclear expression pattern supports mesothelial origin with a diagnostic value like podoplanin (Figure 3). However, its sensitivity is lower than that of D2-40 (variable reported value: 30%, 50–60%) [30,84,85].WT1 and CK5/6: these biomarkers are the least useful in this diagnostic setting, showing the lowest sensitivity (0–45% sensitivity for WT1 and 13–29% sensitivity for CK5/6) and they may be expressed by sarcomatoid carcinomas [45].

Therefore, this differential diagnosis remains a challenge because of morphological overlapping, cytokeratin expression by both tumor types, and the low sensitivity of mesothelial biomarkers. Many recent data have confirmed the usefulness of GATA3 immunoreactivity in the differential diagnosis between PSM and PLC [90,91,92,93]. High rates of GATA3 nuclear expression have been demonstrated in reactive mesothelial proliferation and in MPM, both PEM and PSM, with a heterogeneous expression but usually a high percentage of positive cells [94].

Berg et al. [91] evaluated GATA3 nuclear expression in a subset of PSMs and in a subset of PCLs with spindle cell morphology. All PSMs showed a moderate or strong nuclear expression of GATA3 in at least 1% of neoplastic cells, with a percentage of positive cells greater than 50% in 15 out of 19 cases evaluated. Conversely, most PCLs (84%, 11/13) showed no GATA3 expression and only two cases showed weak intensity of staining in 1% to 25% of tumor nuclei. Furthermore, sarcomatoid carcinomas from sites other than the lung show variable GATA3 positivity. So, their results suggest that GATA3 is a useful biomarker in the differential diagnosis between PSM and PCL and that sarcomatoid neoplasm that does not stain at all for GATA3 is very unlikely to be a mesothelioma [91].

Similar data have been observed by Terra et al. [93]. They demonstrated GATA3 nuclear expression in almost all PSMs (98%, 63/64) while less than half of PCLs (47%, 15/32) were positive, with, among other things, a greater heterogeneity of expression [93]. These data have also been reported by other authors, thus confirming that GATA3 is characterized by a very high diagnostic sensitivity (70–100%) and specificity for PSM diagnosis [92,95,96].

#### 4.1.3. Looking to the Future

In the neoplastic setting, mucin-like membrane proteins are taking on a significant clinical–diagnostic value. Mucin-like membrane proteins are highly O-glycosylated proteins present on the cell surface and these membrane-anchored proteins modified with many glycans showed a good value as neoplasm-related antigens [97]. In this family, sialylated protein HEG homolog 1 (HEG1) is a novel mesothelial-related biomarker, and its expression supports the survival and proliferation of mesothelioma cells [98].

Tsuji et al. [98] were the first to study the potential diagnostic role of HEG1 in MPM, working with the monoclonal antibody (mAb) SKM9-2 obtained by immunizing epithelioid MPM cell lines in mice that is able to recognize the sialylated HEG1. Their data demonstrated a specific immunostaining pattern with an apical membrane staining in neoplastic cells of PEM and epithelioid components of PBM while a weak cytoplasmic expression was detected in PSM, sarcomatoid components of PBM, and PDM. Expression rates of HEG1 were: 98% (89/91) for PEM, 64% (9/14) for PSM, 90% (19/21) for PBM, and 50% (2/4) for PDM, with therefore an overall diagnostic sensitivity of 92% (119/130). This value exceeded those for other studied MPM diagnostic biomarkers (calretinin, WT1, podoplanin, CK5/6, mesothelin). These data demonstrate the high sensitivity of HEG1 expression, but the more interesting finding concerns its diagnostic specificity, showing an insignificant expression of the biomarker in non-mesothelioma tumors and non-neoplastic tissues except in the capillary endothelium and reactive mesothelial cells. In conclusion, Tsuji et al. observed how HEG1 could be an excellent diagnostic biomarker for the pathological diagnosis of MPM, with a diagnostic sensitivity and specificity value of 92% and 99%, respectively.

Further studies have followed to establish the value of HEG1 in this diagnostic setting. Naso et al. [99] performed a large-scale assessment of the SKM9-2 HEG1 antibody and concluded that HEG1 immunohistochemical expression provides a sensitivity comparable to the other mesothelial lineage biomarkers for PEM but it offers a significantly superior specificity. Differently, in the setting of spindle cell neoplasms, HEG1 has shown good diagnostic specificity but poor sensitivity in the differential diagnosis between PSM and sarcomatoid lung carcinomas [99].

Similar results were presented by Churg et al. [100]. In their study, the authors evaluated the diagnostic value of the HEG1–claudin-4 combination in this diagnostic setting. They considered a HEG1 membrane staining as positive and observed a diagnostic sensitivity and specificity of 91% and 99.7%, respectively, for PEM/PBM versus carcinomas from different sites (except for serous carcinoma of the ovary and thyroid carcinomas); at the same time, they further supported the diagnostic role of claudin-4, probably the best broad-spectrum carcinoma biomarker, showing a positive expression in about 1% of MPM. These data are impressive to the point of suggesting how the differential diagnosis between PEM and NSCLC, and probably also metastatic carcinomas from different sites, potentially could be performed with a combination of HEG1 clone SKM9-2 (now commercially available) and claudin-4 immunostaining.

Hiroshima et al. focused on the HEG1 expression pattern both in small biopsies [101] and in cell blocks from pleural effusions [102]. Regarding the small biopsy samples, immunohistochemistry was performed using monoclonal primary anti-HEG1 antibody (clone SKM9-2) and a specific staining score calculated by adding the value of different categories for the staining intensity and the staining extension (percentage of positive cells). The authors interpreted membranous expression in ≥25% neoplastic cells and/or at least a moderate staining intensity as positive. Immunoreactivity for HEG1 was higher in PEMs and in epithelioid components of PBMs than in PSMs and PDMs. In these last two situations, moreover, a HEG1 cytoplasmic expression was observed, without any membranous staining. HEG1 was not expressed in most lung carcinomas, with only rare cases showing positivity and exhibiting focal membranous or cytoplasmic staining with weak to moderate intensity. In addition, the mesothelial population from a significant number of biopsies of fibrous pleuritis showed weak and diffuse HEG1 staining just like a significant number of cases with reactive mesothelial cells showed strong and diffuse HEG1 apical staining; thus, HEG1 showed a low sensitivity in distinguishing sarcomatoid mesothelioma from fibrous pleuritis and a low specificity in distinguishing epithelioid/biphasic mesothelioma from reactive mesothelial proliferations. An interesting finding was that 5.7% of PEMs and 31.2% of PBMs were positive for only one ordinary mesothelial biomarker (WT1, calretinin, or podoplanin) but all of these showed HEG1 expression.

In summary, Hiroshima et al. [101] have reported HEG1 staining in most PEMs and in the epithelioid component of PBMs with a membranous, strong, and diffuse pattern of expression and, at the same time, the substantial lack of expression in lung carcinomas, except for rare cases of focal staining in squamous cell carcinomas or squamous components of lung adenosquamous carcinoma. Therefore, HEG1 has proven to be of excellent value as a mesothelial lineage marker, once again, and could be a useful addition to the current immunohistochemical panel of biomarkers, showing a diagnostic specificity of 92.3% in distinguishing PEM and PBM from all carcinomas and 98.7% in distinguishing PEM and PBM from pulmonary carcinomas.

### 4.2. Is the Mesothelial Proliferation Benign or Malignant in Nature?

One of the most frequent and complex diagnostic problems encountered with pleural biopsies is whether mesothelial proliferation represents a malignant mesothelial neoplasm or a benign reactive hyperplasia [7]. Benign reactive pleural processes, such as reactive mesothelial cell hyperplasia and organizing pleuritis, can clinically and morphologically mimic pleural mesothelioma, showing increased cellularity, entrapped mesothelial cells, and reactive cytologic atypia of the cells (including fibroblasts and endothelial cells).

At the same time, some MPMs can be cytologically bland and may be sampled in minimally invasive areas, making their malignant nature challenging. Although this differential diagnosis is extremely complex, there are some key histopathological features that help resolve the diagnostic question (Table 5) [3,103,104].

Some biomarkers have demonstrated reliability in the differential diagnosis between benign and malignant mesothelial proliferations, with specificity close to 100% and variable sensitivity, and some others are emerging.

#### 4.2.1. Diagnostic Utility of BAP1 Immunohistochemistry

*BRCA*-associated protein 1 (*BAP1*) is a tumor suppressor gene inactivated in approximately 60% of MPMs. *BAP1* mutation is an early event in mesothelioma pathogenesis, and it is one of the most frequently mutated genes in MPM, more often in PEM than PSM, and in MIS [105,106]. Several molecular alterations have been described, such as missense, truncating, and splice site mutations, truncating fusion events, and copy number loss via chr 3p21.1 deletion. It is relevant to note that the loss of BAP1 protein nuclear expression by immunohistochemistry correlates with *BAP1* gene inactivation [107,108], thus BAP1 protein is considered a highly specific biomarker of malignancy in mesothelial proliferations [107,108,109,110].

Bott et al. [111] first reported loss of BAP1 nuclear expression in all their MPMs with *BAP1* loss and/or mutation, but also in some non-mutated cases [111]. Subsequently, Cigognetti et al. [112] reported their exciting results regarding the potential diagnostic role of BAP1 for differentiating mesothelioma from reactive mesothelial proliferations. In the assessment, the authors considered only the nuclear expression of BAP1, and the positivity or negativity status of mesothelial cells was defined as the unambiguous presence or absence of BAP1 expression in mesothelial nuclei, without percentage or intensity cut-off values [112]. The loss of BAP1 nuclear expression showed a diagnostic specificity of 100%, with negative staining only observed in MPMs (145/2128, 66,5%), particularly in PEMs (128/184, 70%) and PBMs (9/15, 60%), while the biomarker was always retained in reactive mesothelial proliferations. Similar data have also been published by other authors [109,113,114,115,116,117,118,119,120,121].

In all the papers, the evaluation of BAP1 status was based exclusively on the presence or absence of protein nuclear expression, regardless of the percentage of positive neoplastic cells in the test and in the absence of a defined cut-off; sometimes the intensity of expression was compared to the inner control to define the results of the test. Therefore, it is reasonable to hypothesize that any percentage of mesothelial cells negative for BAP1 immunohistochemical staining represents a neoplastic clone and is thus diagnostically indicative of malignancy. Loss of BAP1 nuclear expression is regarded as a highly specific indicator of malignancy in mesothelial proliferations (100% specificity for malignancy) (Figure 4), with an accepted and confirmed diagnostic role, although with low diagnostic sensitivity (50–60% sensitivity for MPM) (Figure 5 and Table 6) [3].

#### 4.2.2. Diagnostic Utility of MTAP Immunohistochemistry

The *CDKN2A* gene is located on the chr 9p21 locus near to some other genes, such as *MTAP*. Homozygous deletion of *CDKN2A* is found in approximately 70% of MPMs, with higher prevalence in PSMs (90–100%) than in PEMs and PBMs (40–70%) [122,123]. *CDKN2A* deletions are considered an early step in MPM pathogenesis, also considering that they have been described in some cases of MIS [124,125]. Approximately 75% to 90% of MPMs harboring a *CDKN2A* deletion also exhibit codeletion of the adjacent *MTAP* gene, making its protein product a potential immunohistochemical surrogate for detecting *CDKN2A* deletion [122,126,127]. 5′-Methylthioadenosine phosphorylase (MTAP) plays a critical role in adenosine monophosphate and methionine salvage and its deficiency has been potentially associated with therapies exploiting synthetic lethality [128,129,130]. The first unsatisfactory results regarding the diagnostic value of immunohistochemically detected MTAP deficiency in MPMs [131] have been surpassed by the most recent reported data, which show an excellent specificity and a good sensitivity of the MTAP biomarker, using a monoclonal anti-MTAP primary antibody, for detection of *CDKN2A* homozygous deletion and diagnosis of MPM. In detail, the loss of cytoplasmic MTAP expression by immunohistochemistry (i.e., an entirely negative staining or a lower intensity than the inner positive control) is 65–88% sensitive and 96–100% specific for *CDKN2A* homozygous deletion and 43–65% sensitive (Figure 4) and 100% specific (Figure 5) for MPM diagnosis (Table 6) [113,121,132,133,134,135,136].

In addition, Chapel et al. [135] demonstrated excellent interobserver agreement and interlaboratory reproducibility in the assessment of immunohistochemical MTAP expression, also using different clones (2G4 vs. 42-T). However, although strong evidence suggests that MTAP immunohistochemistry is a reliable surrogate of CDKN2A homozygous deletion and an accurate diagnostic biomarker marker for MPM, further studies are needed to better define the most accurate staining pattern (i.e., staining intensity) and the best cut-off for MTAP staining loss that would allow a definitive diagnosis of MPM [137]. Nevertheless, MTAP is used in some laboratories, and it should already be part of diagnostic immunohistochemical panels in routine practice, especially in complex cases of differential diagnosis between reactive benign mesothelial proliferations and mesothelial malignancy.

### 4.3. New Insights in Immunohistochemical Approach

#### 4.3.1. Diagnostic Utility of Merlin Immunohistochemistry

The *NF2* gene is in chr 22q12 and encodes the Merlin protein, a tumor suppressor acting as an important regulator of contact growth inhibition which mediates suppression of several mitogenic signals and whose inactivation, especially by somatic *NF2* alterations, plays an important role in the pathogenesis of a wide variety of neoplasms, including MPMs [138,139,140,141,142,143].

Although Merlin immunohistochemistry has been studied in the past, the limited specificity and sensitivity of older clones prevented its wider clinical use [144,145]. Recently, a dual immunohistochemical approach for Merlin and YAP1/TAZ, using a newer clone for the former, has provided promising results for the diagnosis of MPM. This approach allows us to predict the status of the *NF2* gene; when the gene is inactivated, an accumulation of YAP1/TAZ occurs [146].

In detail, dual Merlin–YAP/TAZ immunohistochemistry was performed after NGS to find *NF2* gene status and to establish an immunohistochemical cut-off for Merlin based on the value that best discriminated *NF2*-wildtype from *NF2*-altered cases. Protein expression was studied using a specific antibody (Cell Signaling Technology, clone D3S3W) and a cytoplasmic and/or membranous staining pattern was considered as positive. Based on their findings, a complete lack of Merlin expression or a protein expression in less than 10% of tumor cells best correlated with *NF2* alterations detected by NGS. Merlin displayed a good accuracy in distinguishing MPM from benign pleura proliferation, showing a 50% diagnostic sensitivity and a 100% diagnostic specificity, since loss of Merlin expression was seen only in MPMs.

Similar results were subsequently observed by Chapel et al. [136]. Merlin expression was evaluated using a specific antibody (Cell Signaling Technology, clone D1D8) and it was scored as retained (cytoplasmic and/or membranous tumor cell staining) or lost (negative tumor cell staining), irrespective of distribution or intensity. A diagnostic sensitivity and specificity of 52% and 100% were reported, with retained Merlin expression in all 57 reactive mesothelial proliferations. These data indicate that Merlin immunohistochemistry could be a useful tool in the diagnosis of MPM. However, based on the poor correlation between *NF2* molecular status and some aberrant immunohistochemical patterns of Merlin expression, diagnostic emphasis must be placed on complete loss of Merlin expression. Independent studies are needed to validate Merlin immunohistochemistry, both regarding the best antibody clone and the staining conditions as well as the interpretation criteria of the staining, before it can be adopted for routine MPM diagnosis [136,147].

#### 4.3.2. Diagnostic Utility of 5-hmC Immunohistochemistry

5-Hydroxymethylcytosine (5-hmC) is a modified nucleotide derived from 5-methylcytosine through the action of the TET family of DNA hydroxylases [148,149]. A broad variety of neoplasms have shown reduced levels of 5-hmC [150,151] and reduced expression of TET and low levels of 5-hmC have recently been implicated in the tumorigenesis of mesothelioma in rat and human cell lines [152]. However, there is just one preliminary study that investigated only the principle of decreased 5-hmC nuclear expression in MPMs and no work has yet examined its potential application as a diagnostic marker in routine pathology practice [153].

Chapel et al. observed, however, extremely promising results, demonstrating a potentially high diagnostic sensitivity and specificity for MPM diagnosis. In detail, they observed, in a cohort of 49 MPMs (17 PEMs, 22 PBMs, and 10 PSMs) and 23 reactive mesothelial proliferations, that complete loss of 5-hmC nuclear expression (5-hmC antibody, Active Motif, rabbit polyclonal, #39769) in >50% of tumor cells was 100% specific and 92% sensitive for diagnosis of malignancy. This study is a promising step in the validation of 5-hmC immunohistochemistry as a useful diagnostic biomarker for MPM in routine practice, but larger and multi-institutional validation studies are obviously necessary, just like for Merlin.

## 5. Molecular Pathology: What Role?

In the diagnostic workflow of MPM, where the integration of clinical and instrumental data, along with the morphological and immunophenotypic analysis of formalin-fixed and paraffin-embedded samples, remains paramount, a molecular analysis approach can be valuable in cases of challenging interpretation. As previously highlighted, the immunohistochemical loss of cytoplasmic MTAP expression serves as a valuable diagnostic tool in MPM. The MTAP protein is encoded by the *MTAP* gene located at chromosome 9p21, near the *CDKN2A* gene, whose homozygous deletion is observed in a wide range of neoplasms, particularly in MPM, where it occurs in over 70% of cases [154].

For this purpose, in cases where the differential diagnosis lies between a reactive mesothelial proliferation and MPM, with preserved nuclear expression of BAP1 and preserved cytoplasmic expression of MTAP, fluorescence in situ hybridization (FISH) techniques are recommended to investigate the *CDKN2A* status [3].

This is particularly relevant for PSM and PBM, which are more frequently associated with homozygous *CDKN2A* deletions, whereas the epithelioid subtype is more commonly linked to *BAP1* mutations [155]. The study of *CDKN2A* molecular status by FISH should be based on the evaluation of at least 100 neoplastic cells and a cut-off value of 10% should be used for homozygous deletion of 9p21, as reported in the literature [113,121,156,157,158]. Another gene somatically mutated in up to 40% of pleural mesotheliomas is *NF2* [142], which encodes Merlin. This gene is also mutated in the hereditary disorder neurofibromatosis type II, which is associated with the development of neoplasms, particularly in the central and peripheral nervous systems, but not with the onset of mesothelioma. *NF2* mutations are found more frequently in sarcomatoid rather than in epithelioid mesothelioma [159]. *NF2* deficiency has been shown to have utility as a diagnostic tool almost like *CDKN2A* homozygous loss in challenging cases. The use of FISH for the analysis of the *NF2* gene, located on chromosome 22q12, can be valuable in identifying cases of MPM associated with hemizygous *NF2* loss [160] and a recent study showed that the addition of genetic *NF2* screening improved the diagnostic sensitivity and specificity in the diagnosis of MPM [161]. Furthermore, the molecular status of *BAP1*, *CDKN2A*, and *NF2* can be also studied using extractive analyses and sequencing methods to discover inactivating somatic mutations and/or copy-number alterations [154,162]. The methods described so far should not be considered as first-line tools in the routine diagnostic workflow for MPM but rather as techniques aimed at enhancing diagnostic accuracy in challenging cases. The study of *ALK* rearrangements and *EWSR1* fusions is limited to cases associated with atypical disease onset, such as early age at diagnosis.

In addition to the well-known TP53 gene, several other genes have been reported as possibly involved in the pathogenesis of MPM, such as SETDB1, SETD2, and LATS1/2 [163] but, although advanced gene sequencing techniques have achieved a diagnostic sensitivity of 95% [136], there is currently no indication for the routine use of advanced sequencing methods due to the high cost and limited availability of adequately equipped laboratories.

## 6. Conclusions

The diagnosis of pleural disease based on small biopsy samples or, even more so, effusion cytology specimens can be extremely challenging but, at the same time, very fascinating. The diagnosis of MPM almost always requires a multidisciplinary approach, with the morphological features of the lesion acting as the primary guide for diagnostic suspicion. The diagnostic hypothesis should subsequently be confirmed using ancillary methods such as immunohistochemistry and/or molecular tests. A carefully selected immunohistochemical panel, tailored to clinical, radiological, and morphological findings, should be performed. Immunohistochemistry plays a pivotal role in diagnosing mesothelial lesions and relies on the use of several specific biomarkers. The choice is based on multidisciplinary suspicion, bearing in mind that some novel biomarkers have shown to increase the diagnostic accuracy.

## Figures and Tables

**Figure 1 cancers-17-00481-f001:**
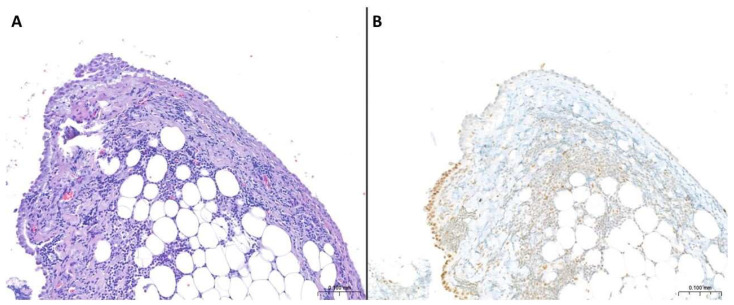
**Mesothelioma in situ.** (**A**) A single-layer surface proliferation of cuboidal mesothelial cells, with mild atypia and only focal nuclear overlapping and pluristratification (H&E staining, original magnification: 10×). (**B**) There is partial immunohistochemical loss of BAP1 nuclear expression, diagnostic of mesothelial malignancy (original magnification: 10×).

**Figure 2 cancers-17-00481-f002:**
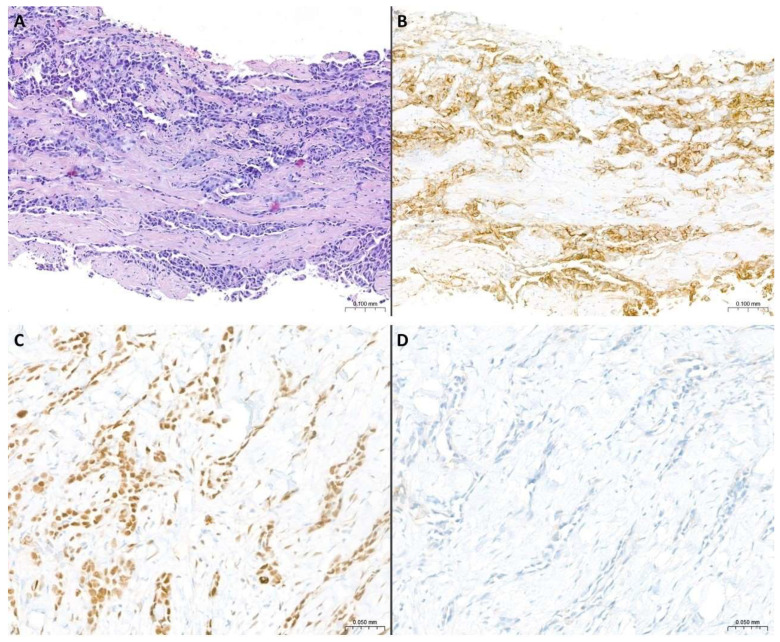
**Morphological and immunohistochemical features of PEM.** (**A**) Pleural biopsy characterized by infiltration from cellular proliferation with a trabecular growth pattern. Mild nuclear atypia and cellular pleomorphism are observed. The stroma appears diffusely collagenized (H&E staining, original magnification: 10×). (**B**) Diffuse immunohistochemical positivity for podoplanin (D2-40) (original magnification: 10×). (**C**) Diffuse immunohistochemical positivity for WT1 (original magnification: 20×). (**D**) Lack of claudin-4 immunohistochemical expression (original magnification: 20×).

**Figure 3 cancers-17-00481-f003:**
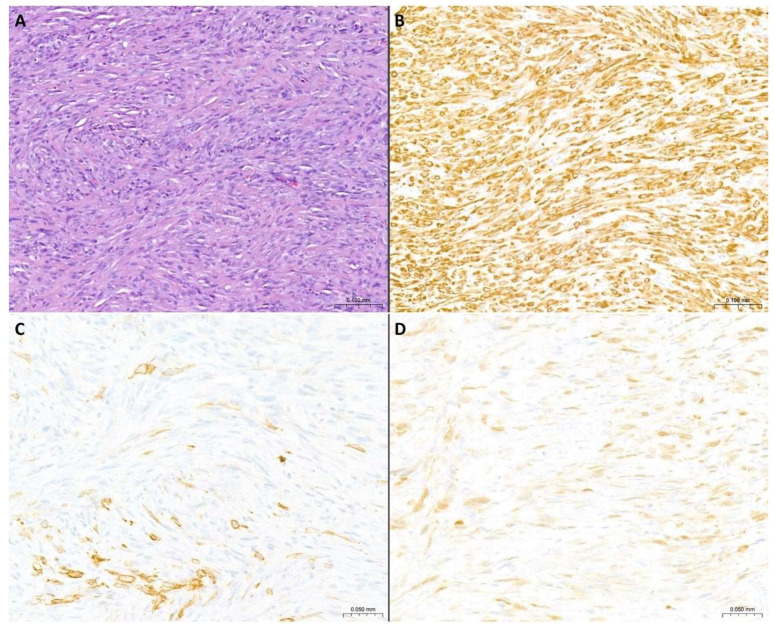
**Morphological and immunohistochemical features of PSM**. (**A**) Pleural biopsy characterized by diffuse infiltration from spindle cell proliferation with a vaguely storiform/small fascicle growth pattern. There are moderate to severe nuclear atypia and evident cellular pleomorphism. Neoplastic proliferation dissects the surrounding sclerotic stroma (H&E staining, original magnification: 10×). (**B**) Diffuse immunohistochemical positivity for pan-cytokeratin (original magnification: 10×). (**C**) Focal immunohistochemical of podoplanin (D2-40) (original magnification: 20×). (**D**) Partial immunohistochemical expression of calretinin with weak intensity (original magnification: 20×).

**Figure 4 cancers-17-00481-f004:**
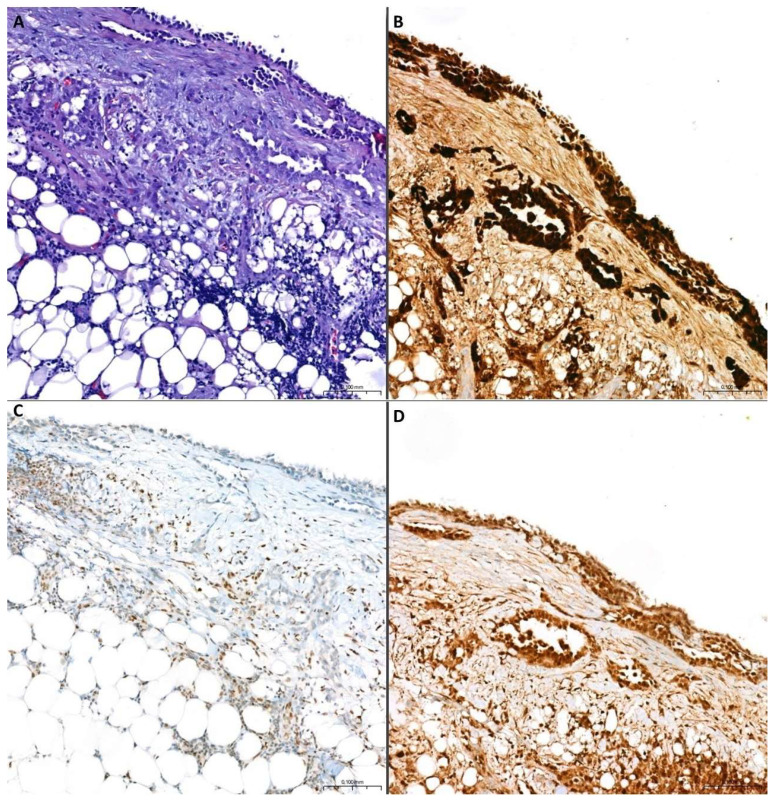
**A case of PEM with diagnostic loss of BAP1 nuclear expression.** (**A**) Pleural biopsy characterized by superficial mesothelial proliferation with a few deep tubular structures near the adipose tissue, without unquestionable features of infiltration (H&E staining, original magnification: 10×). (**B**) Diffuse immunohistochemical positivity for calretinin (original magnification: 10×). (**C**) Loss of immunohistochemical BAP1 nuclear expression in mesothelial cells, diagnostic of malignancy. Positive inner control of surrounding inflammatory cells (original magnification: 10×). (**D**) Retained immunohistochemical MTAP cytoplasmic expression in mesothelial cells (original magnification: 10×).

**Figure 5 cancers-17-00481-f005:**
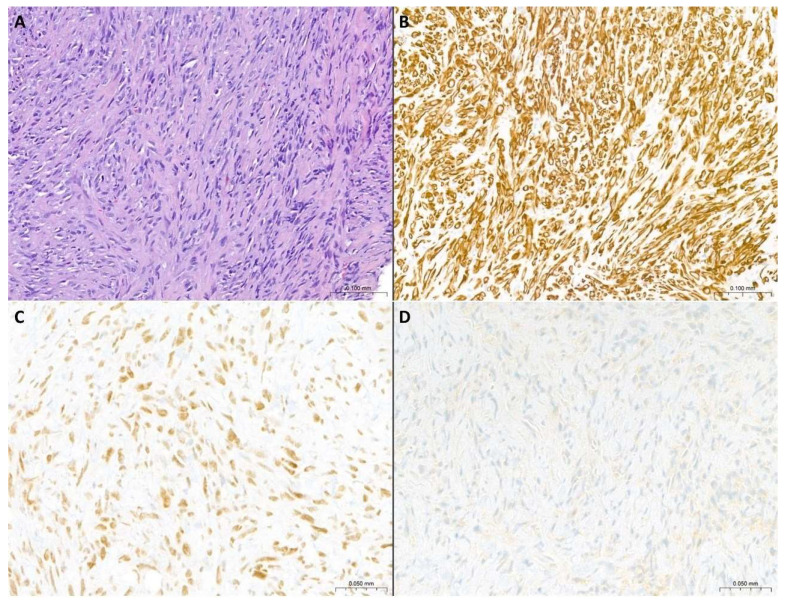
**A case of PSM with diagnostic loss of MTAP cytoplasmic expression.** (**A**) Pleural biopsy characterized by diffuse infiltration from spindle cell proliferation with fascicular growth pattern. There are moderate to severe nuclear atypia and marked cellular pleomorphism (H&E staining, original magnification: 10×). (**B**) Diffuse immunohistochemical positivity for pan-cytokeratin (original magnification: 10×). (**C**) Retained immunohistochemical BAP1 nuclear expression in neoplastic cells (original magnification: 20×). (**D**) Loss of immunohistochemical MTAP cytoplasmic expression in neoplastic cells, feature of malignancy (original magnification: 20×).

**Table 1 cancers-17-00481-t001:** Histopathological features of PEM.

Architectural Patterns	Cytological Features *	Stromal Features
Tubulopapillary	Rhabdoid	Fibrous
Trabecular	Deciduoid	Myxoid
Adenomatoid	Small cell	
Micropapillary	Clear cell	
Solid	Signet ring	
	Lymphohistiocytoid	
	Pleomorphic	

* Excluding PEM conventional cytomorphology.

**Table 2 cancers-17-00481-t002:** Histopathological Grading System for PEM.

**Nuclear Grade**			
**Nuclear Atypia**	**Mild**	**Moderate**	**Severe**
Score	1	2	3
Mitotic Index	Low (<1/2 mm^2^)	Intermediate (2–4/2 mm^2^)	High (≥5/2 mm^2^)
Score	1	2	3
**Nuclear Grade**	**Nuclear Grade I**	**Nuclear Grade II**	**Nuclear Grade III**
Total Score	2 or 3	4 or 5	6
**Necrosis**			
Present		Absent	
**Overall Tumor Grade**			
**Low Grade**		**High Grade**	
Nuclear Grade: I or II Necrosis: Absent		Nuclear Grade: I Necrosis: Present or Nuclear Grade: III Necrosis: Present or Absent	

**Table 3 cancers-17-00481-t003:** Mesothelial neoplasms other than MPM.

Mesothelial Neoplasms	Histological Features	Diagnostic Pitfalls
Adenomatoid tumor	Pseudoglands and pseudovascular spaces. Papillae, tubules, and signet-ring-like spaces. Tumor cells have no atypia and vacuolated cytoplasm, creating a fine network. Lymphoid aggregates often present.	Epithelioid mesothelioma with adenomatoid features
Well-differentiated papillary mesothelial tumor	Surface growth. Papillary architecture with myxoid fibrovascular cores lined by a bland, monotonous monolayer of mesothelial cells. No atypia or stromal invasion. Rare to absent mitoses.	Invasive malignant mesothelioma with papillary features

**Table 4 cancers-17-00481-t004:** Histopathological features of PSM/PDM.

Architectural Patterns	Cytological Features	Stromal Features
Fibrosarcomatous	Lymphohistiocytoid	Desmoplastic
Haphazard	Pleomorphic	With heterologous differentiation *
Patternless	Transitional	

* Osteosarcomatous, rhabdomyosarcomatous, and chondrosarcomatous elements.

**Table 5 cancers-17-00481-t005:** Histopathological features of differential diagnosis between reactive and malignant mesothelial processes.

**Reactive Mesothelial Hyperplasia vs. Malignant Epithelioid Mesothelioma**	
**Reactive Mesothelial Hyperplasia**	**Malignant Epithelioid Mesothelioma**
Absence of stromal invasion Cellularity, even if prominent, is confined to the mesothelial surface/superficial pleural space and decreased in the deep stroma (evidence of “zonation”) Uniform growth and often parallel to pleural surface Loose sheets of cells without stroma No necrosis	Evidence of stromal invasion Dense cellularity, including cells surrounded by stroma, also in deep stroma (sometimes the cellularity is greater in deep stroma than near to pleural surface; absence of “zonation”) Expansile nodules and disorganized growth (cell proliferation grows by infiltrating the stroma, with a typical disorganized arrangement often perpendicular to the surface) Cells surrounded by stroma (“bulky tumor” may involve the mesothelial space without obvious invasion) Presence of tumor necrosis
**Fibrous Pleuritis vs. Sarcomatoid or Desmoplastic Mesothelioma**	
**Fibrous Pleuritis**	**Sarcomatoid or Desmoplastic Mesothelioma**
Absence of stromal invasion Storiform pattern not prominent Cellularity, even if prominent, is confined to the pleural surface and decreased in the deep stroma (evidence of “zonation”) Necrosis, if present, is at the surface and it is often associated with acute inflammation Perpendicularly oriented vessels Uniform thickness of the process	Evidence of stromal invasion Storiform and haphazard pattern prominent Dense cellularity also in deep stroma (sometimes the cellularity is greater in deep stroma than near to pleural surface; absence of “zonation”) Foci of bland necrosis of paucicellular, collagenized tissue Paucity of vessels, without orientation Disorganized growth, with uneven thickness, expansile nodules, and abrupt changes in cellularity

**Table 6 cancers-17-00481-t006:** Immunohistochemical staining pattern and diagnostic sensitivity and specificity of malignant MPM biomarkers.

Biomarker	IHC Pattern	Diagnostic Sensitivity for MPM	Diagnostic Specificity for MPM
BAP1	Loss of nuclear expression	50–60%	100%
MTAP	Loss of cytoplasmic expression	43–65%	96–100%
BAP1 + MTAP	Loss of nuclear expression + Loss of cytoplasmic expression	74–90%	96–100%

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
