# Peer review of "Diagnostic Challenges in the Pathological Approach to Pleural Mesothelioma"

_cancers, 2025, doi:10.3390/cancers17030481_

Round 1
Reviewer 1 Report
Comments and Suggestions for Authors
This is a good overview of the current state of pleural mesothelioma diagnosis. It is comprehensive and logically presented but has too many language issues to consider publishing in its current form. I began listing these but became impatient with the number - early examples include 'combined with...' rather than 'combined to...' on line 59, on line 60-61 the sentence makes no sense, on lines 65-68 the sentence is very hard to follow and understand. There are similar issues with every section (too many to list in detail).
On line 73 the paragraph begins with 'The Materials...' - it is not clear why.
Page 342 includes an unexpected line break.
Apart from the language issues there are some other matters which require attention or explanation:
1. On line 97 the authors describe 'unconventional' PEMS. Are these not just the cytological features described in the WHO blue book?
2. In Table 2 the cells include 'It's not necessary' - is it worth including this column at all?
3. In Table 3 the architectural patterns are not described in the WHO blue book, why include them?
4. The WHO blue book outlines which architectural and cytological features are favourable and unfavourable - would it be helpful to include this information in this review?
5. Figure 2 - Does the WT1 panel show nuclear positivity - if so it is not clear and should be improved
6. CEA is discussed but polyclonal and monoclonal differences not described
7. On line 349 the authors provide a long list of antibodies to be used in the differential of sarcomatoid mesothelioma - are they suggesting all are required in every case?
8. Some information is repeated in several of the sections and this is not necessary
Comments on the Quality of English LanguageThere is considerable scope for improvement (see comments above)
Author Response
Dear Reviewer1,
Thanks for your comments.
Comment 1: This is a good overview of the current state of pleural mesothelioma diagnosis. It is comprehensive and logically presented but has too many language issues to consider publishing in its current form. I began listing these but became impatient with the number - early examples include 'combined with...' rather than 'combined to...' on line 59, on line 60-61 the sentence makes no sense, on lines 65-68 the sentence is very hard to follow and understand. There are similar issues with every section (too many to list in detail).
Response 1: Dear Reviewer, we have corrected the English in the lines indicated and revised the entire article in scientific English.
Comment 2: On line 73 the paragraph begins with 'The Materials...' - it is not clear why.
Response 2: Dear Reviewer, it was an error during the manuscript writing. However, it has been removed
Comment 3: Page 342 includes an unexpected line break.
Response 3: Dear Reviewer, we corrected it.
Apart from the language issues there are some other matters which require attention or explanation:
Comment 4: On line 97 the authors describe 'unconventional' PEMS. Are these not just the cytological features described in the WHO blue book?
Response 4: Dear Reviewer, yes, they are. We have corrected the sentence and added the words 'unconventional cytological features.' These cytomorphological features have been described because they characterize unusual variants of pleural mesothelioma
Comment 5: In Table 2 the cells include 'It's not necessary' - is it worth including this column at all?
Response 5: Dear Reviewer, thank you for the excellent suggestion. We have removed the unnecessary column.
Comment 6: In Table 3 the architectural patterns are not described in the WHO blue book, why include them?
Response 6: Dear Reviewer, we often observe these architectural features in routine practice. Therefore, we thought to include them as they can help in the morphological diagnosis of sarcomatoid/desmoplastic mesothelioma, in our opinion.
Comment 7: The WHO blue book outlines which architectural and cytological features are favourable and unfavourable - would it be helpful to include this information in this review?
Response 7: Dear Reviewer, we considered your suggestion and added the information about the correlation between histological features and biological behavior of the neoplasm.
Comment 8: Figure 2 - Does the WT1 panel show nuclear positivity - if so it is not clear and should be improved
Response 8: Dear Reviewer, we took higher magnification images of WT1 and Claudin-4
Comment 9: CEA is discussed but polyclonal and monoclonal differences not described
Response 9: Dear Reviewer, we have emphasized how anti-CEA monoclonal antibodies are superior to polyclonal antibodies about the diagnostic accuracy
Comment 10: On line 349 the authors provide a long list of antibodies to be used in the differential of sarcomatoid mesothelioma - are they suggesting all are required in every case?
Response 10: Dear Reviewer, thank you for the excellent suggestion. We have emphasized that the choice of the most appropriate panel obviously depends on the morphological features of the lesion. It is an excellent suggestion, as an extensive panel is unnecessary and could also be confusing in diagnostic practice.
Comment 11: Some information is repeated in several of the sections and this is not necessary
Response 11: Dear Reviewer, we have eliminated repetitive sentences.
Best regards
Reviewer 2 Report
Comments and Suggestions for Authors
A very relevant, practically useful and comprehensive updating about the processes conducting to the diagnosis of a malignant pleural mesothelioma (MPM) and the related, manifold diagnostic challenges from the real world.
Three very strong points:
- the overall, full valorization of the histomorphological features of a pleural neoplasm in order to correctly recognize a MPM and to correctly define its subtypes;
- the correct attention getter to the fact that "a history of asbestos exposure should not influence the pathological evalutaion when confirming or excluding mesothelioma" (Lines 55-57);
- the pivotal, conceptual distinction between the question "Is the pleural lesion a mesothelial or non-mesothelial proliferation?" and the question "Is the mesothelial proliferation benign or malignant in nature?".
Just some minor revision needed for the meaning of just a few sentences is unclear respectively at the Lines 73-75, 85-86, 190-191, and 245-246.
Table 1, Column 3 - Ancillary techniques: perhaps "Not necessary" or something similar instead of "It's not necessary".
Reference 75: "Yaziji" instead of "aziji".
Author Response
Dear Reviewer2,
Thanks for your comments.
A very relevant, practically useful and comprehensive updating about the processes conducting to the diagnosis of a malignant pleural mesothelioma (MPM) and the related, manifold diagnostic challenges from the real world.
Comment 1: Three very strong points:
- the overall, full valorization of the histomorphological features of a pleural neoplasm in order to correctly recognize a MPM and to correctly define its subtypes;
- the correct attention getter to the fact that "a history of asbestos exposure should not influence the pathological evalutaion when confirming or excluding mesothelioma" (Lines 55-57);
- the pivotal, conceptual distinction between the question "Is the pleural lesion a mesothelial or non-mesothelial proliferation?" and the question "Is the mesothelial proliferation benign or malignant in nature?".
Comment 2: Just some minor revision needed for the meaning of just a few sentences is unclear respectively at the Lines 73-75, 85-86, 190-191, and 245-246.
Response 2: Dear Reviewer, we corrected all the unclear sentence that you reported
Comment 3: Table 1, Column 3 - Ancillary techniques: perhaps "Not necessary" or something similar instead of "It's not necessary".
Response 3: Dear Reviewer, thank you for suggestion. We have removed the unnecessary column.
Comment 4: Reference 75: "Yaziji" instead of "aziji".
Response 4: Dear Reviewer, we corrected it.
Best regards
Round 2
Reviewer 1 Report
Comments and Suggestions for Authors
Thank you for asking me to look at the revised version of this submission. I am happy with the authors responses and attention to the language. I believe the manuscript is better for them. However there are persisting issues with the language and I have noticed a few other points requiring attention:
1. On line 32 the authors state that histological evaluation is mandatory - I think they later explain that a cytological diagnosis of PEM is possible. Please clarify this contradiction.
2. Line 50 - 'mesothelial cell lining of the...' or 'mesothelial cells lining the...' but not 'mesothelial cells lining of the ...'
3. Line 72. This reference is updated by reference 41.
4. Page 3. The authors do not discuss grading of PEM - this ought to be included
5. Lines 119 and 120. 'you should know them' would be better as 'they should be recognised' and 'they look like' might be better changed to 'mimics'
6. Line 126 - I suggest starting the sentence with Pronounced
7. Line 144, change than to and
8. Line 196 evidents = evidence and then suggests
9. Line 209. The sentence starts According to MIS - this does not make sense
10. Line 233-34. Repeated words
11. Line 231 - again a non-sense sentence
12. Line 286 - 'has' rather than 'it's a biomarker with'
13. Line 290 - usually is not required here
14. Line 362 - 'with' instead of 'when you use a'
15. Line 509-512 Italian rather than English
16. Line 521 - remove 'been'
17. Line 523 - remove 'a'
18. Line 604-5 - Is the Table for all mesotheliomas or just PEMs?
19. Line 643, 'which' rather than 'and'
References - the IMIG guidelines have three versions - are all needed?
Reference 77 - I did not check whether this has been reflected in the text and the subsequent references renumbered.
Comments on the Quality of English Language
Better but still suboptimal in some places
Author Response
Dear Reviewer,
Thanks for the comments and as indicated:
Comment 1. On line 32 the authors state that histological evaluation is mandatory - I think they later explain that a cytological diagnosis of PEM is possible. Please clarify this contradiction.
Response 1. Dear Reviewer, on line 32 we replaced histological with pathological. The diagnosis of mesothelioma typically requires a histological sample. However, it is occasionally possible also in cell block using specific antibodies. They have very high specificity but low sensitivity, therefore the number of false negatives cases in cytology is high. However, in our institution, we always observe pleural effusions sample along with corresponding pleural biopsies in cases suspected of mesothelioma.
Comment 2. Line 50 - 'mesothelial cell lining of the...' or 'mesothelial cells lining the...' but not 'mesothelial cells lining of the ...'
Response 2. Dear Reviewer, we correct it
Comment 3. Line 72. This reference is updated by reference 41.
Response 3. Dear Reviewer, we correct it
Comment 4. Page 3. The authors do not discuss grading of PEM - this ought to be included
Response 4. Dear Reviewer, we included it
Comment 5. Lines 119 and 120. 'you should know them' would be better as 'they should be recognised' and 'they look like' might be better changed to 'mimics'
Response 5. Dear Reviewer, we correct it
Comment 6. Line 126 - I suggest starting the sentence with Pronounced
Response 6. Dear Reviewer, we do it
Comment 7. Line 144, change than to and
Response 7. Dear Reviewer, we correct it
Comment 8. Line 196 evidents = evidence and then suggests
Response 8. Dear Reviewer, we correct it
Comment 9. Line 209. The sentence starts According to MIS - this does not make sense
Response 9. Dear Reviewer, we correct it
Comment 10. Line 233-34. Repeated words
Response 10. Dear Reviewer, we correct it
Comment 11. Line 231 - again a non-sense sentence
Response 11. Dear Reviewer, we correct it
Comment 12. Line 286 - 'has' rather than 'it's a biomarker with'
Response 12. Dear Reviewer, we correct it
Comment 13. Line 290 - usually is not required here
Response 13. Dear Reviewer, we correct it
Comment 14. Line 362 - 'with' instead of 'when you use a'
Response 14. Dear Reviewer, we correct it
Comment 15. Line 509-512 Italian rather than English
Response 15. Dear Reviewer, sorry for the mistake we correct it
Comment 16. Line 521 - remove 'been'
Response 16. Dear Reviewer, we correct it
Comment 17. Line 523 - remove 'a'
Response 17. Dear Reviewer, we correct it
Comment 18. Line 604-5 - Is the Table for all mesotheliomas or just PEMs?
Response 18. Dear Reviewer, the data reported in table 6 are for all mesotheliomas
Comment 19. Line 643, 'which' rather than 'and'
Response 19. Dear Reviewer, we correct it
References - the IMIG guidelines have three versions - are all needed?
Response. Dear Reviewer, we cited only the last update of the IMIG guidelines
Reference 77 - I did not check whether this has been reflected in the text and the subsequent references renumbered.
Best regards